# SAIF: Sparse Adversarial and Imperceptible Attack Framework

## Abstract

Adversarial attacks hamper the decision-making ability of neural networks by perturbing the input signal. The addition of calculated small distortion to images, for instance, can deceive a well-trained image classification network. In this work, we propose a novel attack technique called **S**parse **A**dversarial and **I**mperceptible Attack **F**ramework (SAIF). Specifically, we design imperceptible attacks that contain low-magnitude perturbations at a small number of pixels and leverage these sparse attacks to reveal the vulnerability of classifiers. We use the Frank-Wolfe (conditional gradient) algorithm to simultaneously optimize the attack perturbations for bounded magnitude and sparsity with $O(1/\sqrt{T})$ convergence. Empirical results show that SAIF computes highly imperceptible and interpretable adversarial examples, and largely outperforms state-of-the-art sparse attack methods on ImageNet and CIFAR-10.

## 1 Introduction

Deep neural networks (DNNs) are widely utilized for various tasks such as object detection (Redmon et al., 2016; Girshick, 2015), classification (Krizhevsky et al., 2012; He et al., 2016), and anomaly detection (Chandola et al., 2009). These DNNs are ubiquitously integrated into real-world systems for medical diagnosis, autonomous driving, surveillance, etc., where misguided decision-making can have catastrophic consequences. Therefore, it is crucial to inspect the limitations of DNNs before deployment in such safety-critical systems.

Adversarial attacks (Szegedy et al., 2014) are one means of exposing the fragility of DNNs. In the classification task, these attacks can fool well-trained classifiers to make arbitrary (untargeted) (Moosavi-Dezfooli et al., 2016) or targeted misclassifications (Carlini & Wagner, 2017) by negligibly manipulating the input signal. For instance, a road sign classifier can be led to interpret a slightly modified stop sign as a speed limit sign (Benz et al., 2020). Such adversarial attacks fool learning algorithms with high confidence while being imperceptible to the human eye. Most attack methods achieve this by constraining the pixel-wise magnitude of the perturbation. Minimizing the number of modified pixels is another strategy for making

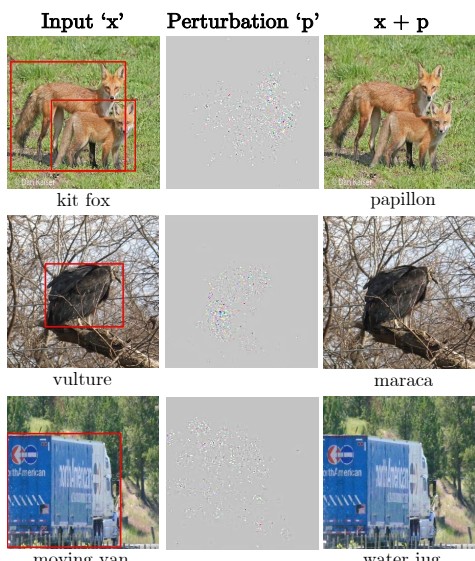

Figure 1: Using the Frank-Wolfe algorithm to jointly constrain the perturbation magnitude and sparsity, we craft an adversarial attack that is very sparse and highly imperceptible. By restricting attack sparsity, we can visualize the most vulnerable pixels in an image. The GT bounding boxes for the subject of the input **x** are drawn in red. Note that SAIF mostly distorts pixels within that region. Inception-v3 is used for predicting labels.

the perturbations unnoticeable (Su et al., 2019; Narodytska & Kasiviswanathan, 2017). Bringing these two together generates high-stealth attacks (Modas et al., 2019; Croce & Hein, 2019; Dong et al., 2020; Fan et al., 2020; Williams & Li, 2023).

Existing methods, however, fail to produce attacks with simultaneously *very* high sparsity and low magnitude perturbations. In this paper, we address this limitation by designing a novel method that produces strong adversarial attacks with a significantly low perturbation strength and high sparsity. Our proposed approach, we call **S**parse **A**dversarial and **I**mperceptible attack **F**ramework (SAIF), minimally modifies only a fraction of pixels to generate highly concealed adversarial attacks.

SAIF aims to jointly minimize the perturbation magnitude and sparsity. We formulate this objective as a constrained optimization problem. Previous works propose projection-based methods (such as PGD (Madry et al., 2018)) to optimize similar objectives, however, these require a projection step at each iteration to obtain feasible solutions (Croce & Hein, 2019). Such projections give rise to iterates very close to/at the constraint boundary, and projecting the solutions can diminish their 'optimality'. Optimization methods such as ADMM (Xu et al., 2019; Fan et al., 2020) and homotopy (Fan et al., 2020) have also been explored, but have prohibitively long running times for large images.

To address these limitations, we propose to leverage the Frank-Wolfe algorithm (FW) (Frank et al., 1956) to optimize our objective. FW is a projection-free, iterative method for solving constrained convex optimization problems using conditional gradients. In contrast to PGD attacks, the absence of a projection step allows FW to find perturbations well within the constraint boundaries. Throughout optimization, the iterates are within the constraint limits since they are convex combinations of feasible points. Moreover, there are several algorithmic variants of FW for efficient optimization.

Furthermore, the benefit from adversarial examples can be maximized by examining the vulnerabilities of deep networks alongside model explanations (Ignatiev et al., 2019; Wang et al., 2022; Xu et al., 2019). Magnitude-constrained attacks distort all image pixels, leaving little room to interpret the additive perturbations. Our proposed attack offers explicit control over distortion magnitude and sparsity. This facilitates more controlled and straightforward semantic analyses, such as identifying the top-'$k$' pixels that are critical for fooling DNNs.

Concretely, the contributions of this paper are:

- We introduce a novel optimization-based adversarial attack that is visually imperceptible due to low-magnitude distortions to a fraction of image pixels.
- We show through comprehensive experiments that, for tight sparsity and magnitude constraints, SAIF outperforms state-of-the-art sparse attacks by a large margin (by $\geq 2\times$ higher fooling rates for most thresholds).
- Our sparse attack provides transparency by indicating vulnerable pixels in natural images. By quantitative evaluation of the overlap between perturbations and salient image regions, we emphasize the utility of SAIF for such analyses.

## 2 RELATED WORKS

**Magnitude-Constrained Adversarial Attacks.** The first discovered adversarial attack by Szegedy et al. (2014) uses box-constrained L-BFGS to minimize the $\ell_2$ norm of additive distortion, however, it is slow and does not scale to larger inputs. To overcome speed limitations, the Fast Gradient Sign Method (FGSM) (Goodfellow et al., 2015) uses $\ell_\infty$ constrained gradient ascent w.r.t. the loss gradient for each pixel, to compute an efficient attack but with poor convergence. Projected Gradient Descent (PGD) is another optimization-based attack algorithm that is fast and computationally cheap, but yields solutions closer to the boundary and often fails to converge (Madry et al., 2018). Auto-attack (Croce & Hein, 2020) addresses the convergence limitations of PGD. Nevertheless, these attacks distort all the image pixels, potentially leading to high visual perceptibility. We address this shortcoming by explicitly constraining the sparsity of the adversarial perturbations.

**Sparsity-Constrained Adversarial Attacks.** The Jacobian-based Saliency Map Attack (JSMA) denotes pixel-wise saliency by backpropagated gradient magnitudes, then searches over the most salient pixels for a sparse targeted perturbation (Papernot et al., 2016). This attack is slow and fails to scale to larger images. SparseFool (Modas et al., 2019) extends DeepFool (Moosavi-Dezfooli et al., 2016) to a sparse attack within the valid pixel magnitude bounds. The attack, however, is untargeted. Croce & Hein (2019) devise a black-box attack by evaluating the impact of each pixel on logits and randomly sampling salient pixels to find a feasible sparse combination. Similar to JSMA, these attacks are expensive to compute, visually noticeable, and unstructured. The same

limitations hold for SA-MOO (Williams & Li, 2023). StrAttack (Xu et al., 2019) uses ADMM (Alternating Direction Method of Multipliers) to optimize for group sparsity and perturbation magnitude constraints, but the perturbations are computationally expensive, visually noticeable, and of low sparsity. SAPF (Fan et al., 2020) also uses ADMM with projections to solve a factorized objective. It fails to converge for a tighter sparsity budget, requires extensive hyperparameter tuning, and has a prohibitively long running time. Among generator-based methods, Dong et al. (2020) propose GreedyFool, a two-stage approach to greedily sparsify perturbations obtained from a generator. Similarly, TSAA (He et al., 2022) generates sparse, magnitude-constrained adversarial attacks with high black-box transferability. The perturbations are typically spatially contiguous and are therefore more noticeable than other sparse attacks. The design of our attack, and employing Frank-Wolfe for optimization, yield highly sparse and inconspicuous adversarial examples efficiently.

**Understanding Adversarial Attacks**   A fairly novel research direction examines adversarial examples and model explanations in conjunction, by noting the overlap between core ideas in the domains. On simple datasets such as MNIST, (Ignatiev et al., 2019) demonstrates a hitting set duality between model explanations and adversarial examples. Similarly, Wang et al. (2022) leverage adversarial attacks to devise a novel model explainer. Xu et al. (2019) examine the correspondence of attack perturbations with discriminative image features. We formulate our attack with an explicit sparsity constraint which emphasizes only the most vulnerable pixels in an image. We also empirically analyze the overlap of adversarial perturbations and salient regions in images.

## 3 BACKGROUND

In this section, we introduce the notations and conventions for adversarial attacks. We also provide a brief review of the Frank-Wolfe algorithm.

### 3.1 ADVERSARIAL ATTACKS

Given an image $\mathbf{x} \in \mathbb{R}^{h \times w \times c}$, a trained classifier $f : \mathbb{R}^{h \times w \times c} \to \{1 \ldots k\}$ that maps the image to one of $k$ classes, and $f(\mathbf{x}) = c$. Adversarial attacks aim at finding $\mathbf{x}'$ that is very similar to $\mathbf{x}$ by a distance metric, i.e. $||\mathbf{x} - \mathbf{x}'||_p \leq \epsilon$, $(p \in \{0, 1, 2, \infty\}, \epsilon$ is small) such that $f(\mathbf{x}') = t$, where $t \neq c$.

Depending on the adversary's knowledge of the target model, adversarial attacks can be white-box (known model architecture and parameters) or black-box (unknown learning algorithm, the attacker only sees the most likely prediction given an input). We adopt the white-box setting in this work, however, gradient approximations (Chen et al., 2020) can be used for black-box attacks.

### 3.2 FRANK WOLFE ALGORITHM

The Frank-Wolfe algorithm (FW) (Frank et al., 1956) is a first-order, projection-free algorithm for optimizing a convex function $f(\mathbf{x})$ over a convex set $\mathbf{X}$ (Algorithm 2, Appendix A). The key advantage of FW is that the iterates $\mathbf{x}_t$ always remain feasible $(\mathbf{x}_t \in \mathbf{X})$ throughout the optimization process. The algorithm was popularized for machine learning applications by Jaggi (2013) with rigorous proofs in objective value $f(\mathbf{x}_t) - f(\mathbf{x}^*)$, where $\mathbf{x}^*$ is the optimal point.

FW is a projection-free method since it solves a linear approximation of the objective over $\mathbf{X}$ (see step 3 in Algorithm 2, Appendix A), known as the Linear Minimization Oracle (LMO). For convex optimization, the optimality gap $f(\mathbf{x}_t) - f(\mathbf{x}^*)$ is upper bounded by the duality gap $g(\mathbf{x}_t) = \min_{a \in \mathcal{A}} \langle \nabla f(\mathbf{x}_t), a \rangle$, which measures the instantaneous expected decrease in the objective and converges at a sub-linear rate of $O(1/T)$ for Algorithm 2 (Jaggi, 2013). FW can locally solve non-convex objectives over convex regions with $O(1/\sqrt{T})$ convergence (Lacoste-Julien, 2016).

The first work using Frank-Wolfe for adversarial attacks Chen et al. (2020) constrains only the magnitude of perturbation $||\mathbf{x} - \mathbf{x}'||_\infty$. As a result, the crafted attack is non-sparse. Recent works also employ Frank-Wolfe for explaining predictions (Roberts & Tsiligkaridis, 2021) and for faster adversarial training (Tsiligkaridis & Roberts, 2022; Wang et al., 2019).

Our motivation to employ FW for optimizing SAIF is twofold: **(1)** it is a 'conservative' algorithm with iterates strictly in the feasible region throughout the optimization. Its projection-free nature prevents sub-optimal solutions common in methods like PGD, and **(2)** it has sparsity-inducing properties, which fits our goal.

## 4 METHOD

Our goal is to calculate an adversarial attack that has low magnitude and high sparsity simultaneously. Formally, the perturbations should have low $\ell_0$-norm and low $\ell_\infty$-norm to satisfy the sparsity and magnitude requirements, respectively. Moreover, the (untargeted) attack should maximize classification loss for the true class.

To implement such an attack, we define a sparsity-constrained mask $\mathbf{s}$ to preserve pixels of an additive adversarial perturbation $\mathbf{p}$. We also impose an $\ell_\infty$ constraint on the magnitude of $\mathbf{p}$. Decoupling the attack into a sparse mask and perturbation also allows visualizing vulnerable pixels of the image.

**Untargeted Attack.** Given $f(\mathbf{x}) = c$, we define our untargeted objective function $D(\mathbf{s}, \mathbf{p})_{adv}$ as

$$D(\mathbf{s}, \mathbf{p})_{adv} = \Phi(\mathbf{x} + \mathbf{s} \odot \mathbf{p}, c) \tag{1}$$

Here $\Phi(., c)$ is the classification loss function, (e.g., cross-entropy), with respect to the true class $c$.

Then, the optimization objective is to maximize the loss for the original class as:

$$\max_{\mathbf{s},\mathbf{p}} D(\mathbf{s}, \mathbf{p})_{adv}, \ \text{s.t.} \|\mathbf{s}\|_1 \leq k, \ \mathbf{s} \in [0,1]^{h \times w \times c}, \|\mathbf{p}\|_\infty \leq \epsilon \tag{2}$$

This formulation not only highlights the vulnerable regions of the image to perturb via $\mathbf{s}$, but also yields an adversarial attack method where we can explicitly control the sparsity using $k$ and the perturbation magnitude per pixel with $\epsilon$. Note that following the common practice to simplify optimization over the NP-hard $\ell_0$ constraint, we use $\ell_1$ as a convex surrogate for $\ell_0$ over $\mathbf{s}$ (Macdonald et al., 2022; He et al., 2022).

**Targeted Attack.** We extend (1) to targeted attacks by replacing $c$ with a chosen target class $\tilde{c}$.

$$D(\mathbf{s}, \mathbf{p})_{\tilde{c},adv} = \Phi(\mathbf{x} + \mathbf{s} \odot \mathbf{p}, \tilde{c}), \quad \tilde{c} \neq c \tag{3}$$

Then to enhance the odds of predicting $\tilde{c}$, we minimize $D(\mathbf{s})_{\tilde{c},adv}$ to obtain the SAIF attack:

$$\min_{\mathbf{s},\mathbf{p}} D(\mathbf{s}, \mathbf{p})_{\tilde{c},adv}, \ \text{s.t.} \|\mathbf{s}\|_1 \leq k, \ \mathbf{s} \in [0,1]^{h \times w \times c}, \|\mathbf{p}\|_\infty \leq \epsilon \tag{4}$$

**Optimization.** We use Frank-Wolfe as the solver for our objectives 2 and 4 in order to ensure that the variable iterates remain feasible (see Algorithm 1). The algorithm proceeds by moving the iterates towards a minimum by simultaneously minimizing the objective w.r.t. $\mathbf{s}$ and $\mathbf{p}$.

To constrain $\mathbf{s}$ we use a non-negative $k$-sparse polytope, which is a convex hull of the set of vectors in $[0,1]^{h \times w \times c}$, each vector admitting at most $k$ non-zero elements. We adopt the method in Macdonald et al. (2022) to perform the LMO over this polytope. That is, for $\mathbf{z_t}$ we choose the vector with at most $k$ non-zero entries, where the conditional gradient $\mathbf{m}_t^s$ assumes $k$ smallest negative values (thus highest in magnitude). These $k$ components of $\mathbf{z_t}$ are then set to 1 and the rest to zero. For example, if $k = 10$ and there are 20 negative values in $\mathbf{m}_t^s$, the 10 smallest values are set to 1 and the rest to 0.

For $\mathbf{p}$, the LMO of $\ell_\infty$ has a closed-form solution (Chen et al., 2020).

$$\mathbf{v}_t = -\epsilon \cdot \text{sign}(\mathbf{m}_t^p) + \mathbf{x} \tag{5}$$

Note that it is possible to combine $\mathbf{s}$ and $\mathbf{p}$ into one variable using a method such as Gidel et al. (2018). However, in doing so we would lose the interpretability brought by disentangling the sparse mask $\mathbf{s}$. This is because enforcing an $\ell_1$ constraint is not the same as the currently enforced $k$-sparse polytope. Therefore, such a dual constraint would result in a perturbation of varying values which is harder to directly interpret than a $[0, 1]$-valued mask.

Since the objective of SAIF is non-convex, a monotonicity guarantee is helpful to ensure that the separate optimizations of each variable sync well. To this end, we use the following adaptive step size formulation (Carderera et al., 2021; Macdonald et al., 2022) for monotonicity in the objective:

$$\eta_t = \frac{1}{2^{r_t}\sqrt{t+1}} \tag{6}$$

where we choose the $r_t \in \mathbb{N}$ by increasing from $r_{t-1}$, until we observe primal progress of the iterates. This method is conceptually similar to the backtracking line search technique often used with standard gradient descent.

---

**Algorithm 1:** SAIF - Adversarial attack using Frank-Wolfe for joint optimization.

---

**Input:** Clean image $\mathbf{x} \in [0, I_{max}]^{h \times w \times c}$, target class $\tilde{c} \in 1 \ldots k$,
$\quad \mathbf{s}_0 \in \mathcal{C}_s = \{\mathbf{s} \in [0, 1]^{h \times w \times c} : \|\mathbf{s}\|_1 \leq k\}$,
$\quad \mathbf{p}_0 \in \mathcal{C}_p = \{\mathbf{p} \in [0, I_{max}]^{h \times w \times c} : \|\mathbf{p}\|_\infty \leq \epsilon\}$.
**Output:** Perturbation $\mathbf{p}$, Sparse mask $\mathbf{s}$

1 **for** $t = 1, \ldots, T$ **do**
2 $\quad \mathbf{m}_t^p = \nabla_p D(\mathbf{s}_{t-1}, \mathbf{p}_{t-1})$
3 $\quad \mathbf{m}_t^s = \nabla_s D(\mathbf{s}_{t-1}, \mathbf{p}_{t-1})$
4 $\quad \mathbf{v}_t = \mathrm{argmin}_{\mathbf{v} \in \mathcal{C}_p} \langle \mathbf{m}_t^p, \mathbf{v} \rangle$
5 $\quad \mathbf{z}_t = \mathrm{argmin}_{\mathbf{z} \in \mathcal{C}_s} \langle \mathbf{m}_t^s, \mathbf{z} \rangle$
6 $\quad \mathbf{p}_t = \mathbf{p}_{t-1} + \eta_t(\mathbf{v}_t - \mathbf{p}_{t-1})$
7 $\quad \mathbf{s}_t = \mathbf{s}_{t-1} + \eta_t(\mathbf{z}_t - \mathbf{s}_{t-1})$
8 **end**

---

## 5 EXPERIMENTS

We evaluate SAIF against several existing methods for both targeted and untargeted attacks. We report performance on the effectiveness as well as saliency of adversarial attacks.

**Dataset and Models** We use the ImageNet classification dataset (ILSVRC2012) (Krizhevsky et al., 2012) in our experiments, which has $[299 \times 299]$ RGB images belonging to 1,000 classes. We evaluate all attacks on 5,000 samples chosen from the validation set. For classification, we test on two deep convolutional neural network architectures, namely Inception-v3 (top-1 accuracy: 77.9%) and ResNet-50 (top-1 accuracy: 74.9%). We use the pre-trained models from Keras applications (Chollet et al., 2015). We also report results on CIFAR-10 with VGG-16 as the classifier.

**Implementation** We implement the experiments in Julia and use the Frank-Wolfe variants library (Besançon et al., 2021). We code the classifier and gradient computation backend in Python using TensorFlow and Keras deep learning frameworks. The experiments are run on a single Tesla V100 SXM2 GPU, for an empirically chosen number of iterations $T$ for each dataset. SAIF typically converges in $\sim$20 iterations, however, we relax the maximum iterations to $T = 100$ in our experiments.

### 5.1 EVALUATION METRICS

**Adversarial Attacks.** Adversarial attacks are commonly evaluated by the attack success rate.

- **Attack Success Rate (ASR).** A targeted adversarial attack is deemed successful if perturbing the image fools the classifier into labeling it with a premeditated target class $\tilde{c}$. An untargeted attack is successful if it leads the classifier to predict *any* incorrect class. Given $n$ images in a dataset, if $m$ attacks are successful, the attack success rate is defined as ASR $= m/n(\%)$.

Note that for RGB images, SparseFool (Modas et al., 2019) and Greedyfool (Dong et al., 2020) average the perturbation $\mathbf{p}$ across the channels and report the ASR for sparsity $\|\mathbf{p}_{\mathrm{flat}}\|_0 \leq k$, where $\mathbf{p}_{\mathrm{flat}} \in [0, I_{max}]^{h \times w}$. Since for SAIF the sparsity constraint $k$ applies across all $h \times w \times c$ pixels, we report the ASR for all methods **without** averaging the final perturbation across the channels (i.e. for $\|\mathbf{p}\|_0 \leq k, \mathbf{p} \in [0, I_{max}]^{h \times w \times c}$).

**Attack Saliency.** We use the following metric to capture the correspondence between the vulnerable and salient pixels in the input images. The score represents the overlap of the ground-truth (GT) bounding box of the subject of the input image with the (sparse) adversarial perturbation.

- **Localization (Loc.) (Chattopadhay et al., 2018)** Effectively the same as IoU for object detection. Given image pixels X, GT salient pixels S and SAIF sparse mask A, the localization score is:

$$\mathrm{Loc.} = \frac{\|A \cap S\|_0}{\|S\|_0 + \|A \cap (X \setminus S)\|_0} \tag{7}$$

In the event of perfect correspondence between the GT salient regions and adversarial perturbation, Loc.$\rightarrow$1. Whereas, Loc.$\rightarrow$0 when there is poor overlap between the two.

| $\|\mathbf{p}\|_\infty \le \epsilon$ | Attacks | Inception-v3 | | | | | ResNet50 | | | | |
|---|---|---|---|---|---|---|---|---|---|---|---|
| | | Sparsity '$k$' | | | | | Sparsity '$k$' | | | | |
| | | 100 | 200 | 600 | 1000 | 2000 | 100 | 200 | 600 | 1000 | 2000 |
| $\epsilon = 255$ | GreedyFool | 0.40 | 1.19 | 19.36 | 49.70 | 87.82 | 0.59 | 2.59 | 28.94 | 70.06 | 96.21 |
| | TSAA | 0.00 | 1.15 | 31.61 | 77.01 | 100.0 | -* | -* | -* | -* | -* |
| | Homotopy-Attack | 0.00 | 18.23 | 90.97 | 100.0 | 100.0 | 0.00 | 0.00 | 0.00 | 0.00 | 0.00 |
| | SAIF (Ours) | **55.97** | **84.00** | **100.0** | **100.0** | **100.0** | **80.10** | **100.0** | **100.0** | **100.0** | **100.0** |
| | | 100 | 200 | 600 | 1000 | 2000 | 100 | 200 | 600 | 1000 | 2000 |
| $\epsilon = 100$ | GreedyFool | 0.20 | 0.40 | 4.59 | 12.18 | 25.35 | 0.20 | 1.59 | 12.57 | 28.54 | 44.11 |
| | Homotopy-Attack | 9.04 | 18.27 | 72.07 | **100.0** | **100.0** | 0.00 | 0.00 | 0.00 | 0.00 | 0.00 |
| | SAIF (Ours) | **21.73** | **66.27** | **100.0** | **100.0** | **100.0** | **59.01** | **90.72** | **100.0** | **100.0** | **100.0** |
| | | 1000 | 2000 | 3000 | 4000 | 5000 | 1000 | 2000 | 3000 | 4000 | 5000 |
| $\epsilon = 10$ | GreedyFool | 0.20 | 1.39 | 2.99 | 5.59 | 8.98 | 5.59 | 18.36 | 30.14 | 39.12 | 47.50 |
| | Homotopy-Attack | 0.00 | 30.05 | 58.27 | 69.59 | 85.03 | 0.00 | 0.00 | 0.00 | 0.00 | 0.00 |
| | SAIF (Ours) | **14.28** | **44.79** | **90.29** | **94.97** | **100.0** | **60.19** | **80.02** | **89.93** | **90.42** | **100.0** |

Table 1: Quantitative evaluation of **targeted** attack on ImageNet. We report the ASR for varying constraints on sparsity '$k$' and $\ell_\infty$-norm of the magnitude of perturbation '$\epsilon$'. (* TSAA codebase lacks pre-trained generators for targeted attacks on ResNet50 and lower $\epsilon$.)

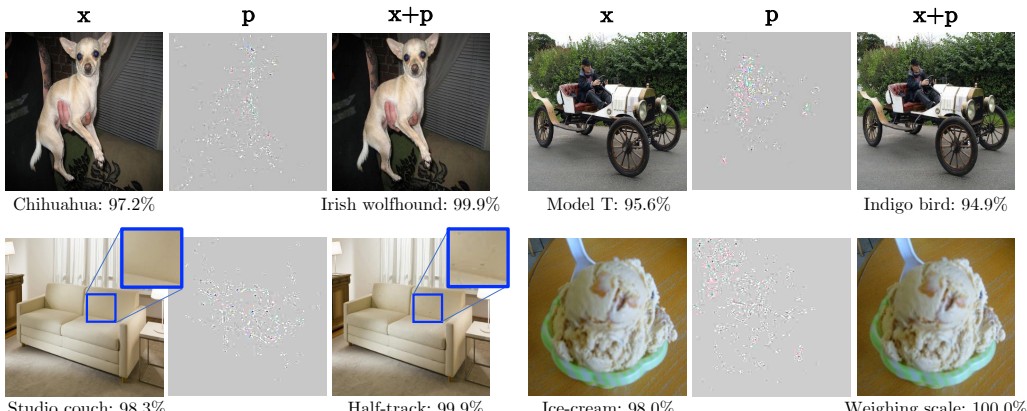

Figure 2: Qualitative results of **targeted** SAIF attack on Inception-v3 trained on ImageNet, using $\epsilon = 100$ (39% of the dynamic range of image) and $k = 400$ (0.15% of pixels). The source and target class, along with the corresponding probability, are stated below each $\mathbf{x}$ and $\mathbf{x} + \mathbf{p}$ respectively.

## 6 RESULTS

**Quantitative Results.** We evaluate the ASR of all attack methods on a range of constraints on perturbation magnitude $\epsilon$ and sparsity $k$. The results for *targeted* attacks on ImageNet are reported in Table 1. The target class is randomly chosen for each sample. Note that TSAA (He et al., 2022) does not evaluate attacks for $\epsilon = 100$. Moreover, for both targeted and untargeted attacks, SAPF (Fan et al., 2020) fails for all the evaluated thresholds.

Table 1 demonstrates that SAIF consistently outperforms both targeted attack baselines by a large margin for all $\epsilon$ and $k$ thresholds. For smaller $\epsilon$, GreedyFool fails to attack samples unless the sparsity threshold is significantly relaxed. A similar pattern is observed for the sparsity budget - competing attacks completely fail for tighter bounds on perturbation magnitude (see $\epsilon = 10/100/255$ at $k = 2000$). Moreover, at lower $k$, ResNet50 is easier to fool than Inception-v3.

We also compare the ASR for *untargeted* attacks against the baselines in Table 2. Here as well SAIF significantly outperforms other attacks, particularly on tighter perturbation bounds. By jointly optimizing over the two constraints, we are able to fool DNNs with extremely small distortions. For instance, at $\epsilon = 10$, SAIF modifies only 0.37% pixels per image to successfully perturb 89.02% of input samples. This is more than twice the ASR of state-of-the-art sparse attack methods. Similarly, at $\epsilon = 255$, only 0.03% pixels are attacked to achieve a perfect ASR. On CIFAR-10, SAIF achieves $\sim$3$\times$ higher ASR on lower sparsity thresholds. The results are reported in Appendix C.

Note that sparse attacks like ours address a more challenging problem than $\ell_\infty$-norm threat models since both the perturbation magnitude **and** sparsity are to be constrained. For $\ell_\infty$ constraints $\epsilon =$ 10, 100, and 255, AutoAttack (Croce & Hein, 2020) achieves 100% ASR with a perturbation sparsity

| $\|\mathbf{p}\|_\infty \leq \epsilon$ | Attacks | Inception-v3 | | | | | ResNet50 | | | | |
|---|---|---|---|---|---|---|---|---|---|---|---|
| | | Sparsity '$k$' | | | | | Sparsity '$k$' | | | | |
| | | 10 | 20 | 50 | 100 | 200 | 10 | 20 | 50 | 100 | 200 |
| $\epsilon = 255$ | SparseFool | 1.59 | 4.39 | 15.37 | 32.14 | 32.14 | 1.79 | 3.19 | 8.78 | 17.76 | 33.33 |
| | GreedyFool | 3.99 | 7.58 | 16.57 | 35.33 | 62.87 | 4.19 | 7.78 | 21.76 | 42.12 | 72.26 |
| | TSAA | 0.00 | 0.00 | 0.00 | 0.00 | 2.02 | 0.00 | 0.00 | 0.00 | 1.95 | 16.06 |
| | PGD $\ell_0 + \ell_\infty$* | 0.81 | 0.81 | 3.63 | 5.65 | 7.80 | 11.20 | 11.20 | 11.67 | 12.25 | 12.25 |
| | SA-MOO* | 9.52 | 10.04 | 14.28 | 38.09 | 39.47 | 27.98 | 44.32 | 45.12 | 54.83 | 60.91 |
| | SAIF (Ours) | **19.88** | **60.16** | **90.05** | **100.0** | **100.0** | **38.25** | **61.72** | **100.0** | **100.0** | **100.0** |
| | | 10 | 20 | 50 | 100 | 200 | 10 | 20 | 50 | 100 | 200 |
| $\epsilon = 100$ | SparseFool | 0.79 | 3.39 | 9.98 | 27.54 | 48.90 | 1.39 | 2.59 | 7.58 | 19.56 | 35.72 |
| | GreedyFool | 2.39 | 3.79 | 10.18 | 23.15 | 45.11 | 2.20 | 5.99 | 15.77 | 34.73 | 61.68 |
| | PGD $\ell_0 + \ell_\infty$* | 0.00 | 1.24 | 3.29 | 5.22 | 6.86 | 11.67 | 12.25 | 12.25 | 12.25 | 14.49 |
| | SA-MOO* | **4.67** | 9.52 | 9.98 | 14.28 | 23.81 | **29.86** | 34.10 | 35.56 | 39.83 | 50.24 |
| | SAIF (Ours) | 0.00 | **28.91** | **60.26** | **90.03** | **100.0** | 20.42 | **41.26** | **79.32** | **100.0** | **100.0** |
| | | 200 | 500 | 1000 | 2000 | 3000 | 200 | 500 | 1000 | 2000 | 3000 |
| $\epsilon = 10$ | SparseFool | 4.19 | 14.17 | 38.92 | 67.86 | 82.24 | 11.98 | 39.92 | 69.46 | 90.02 | 95.61 |
| | GreedyFool | 8.78 | 22.55 | 40.52 | 65.07 | 77.25 | 18.77 | 47.70 | 74.65 | 93.41 | 97.21 |
| | TSAA | 0.00 | 0.00 | 0.00 | 0.00 | 0.00 | 0.00 | 0.00 | 0.00 | 0.59 | 6.89 |
| | PGD $\ell_0 + \ell_\infty$* | 4.83 | 7.69 | 11.21 | 22.03 | 30.47 | 11.28 | 11.28 | 11.64 | 12.80 | 14.83 |
| | SA-MOO* | 4.76 | 4.98 | 5.02 | 9.52 | 10.98 | 15.56 | 17.82 | 18.01 | 18.94 | 20.97 |
| | SAIF (Ours) | **10.21** | **52.40** | **89.02** | **90.00** | **100.0** | **50.04** | **73.09** | **95.00** | **100.0** | **100.0** |

Table 2: Quantitative evaluation of **untargeted** attack on Inception-v3 and ResNet50 trained on ImageNet dataset. We report the ASR for varying constraints on sparsity '$k$' and $\ell_\infty$-norm of the magnitude of perturbation '$\epsilon$'. Black-box attacks are marked with *.

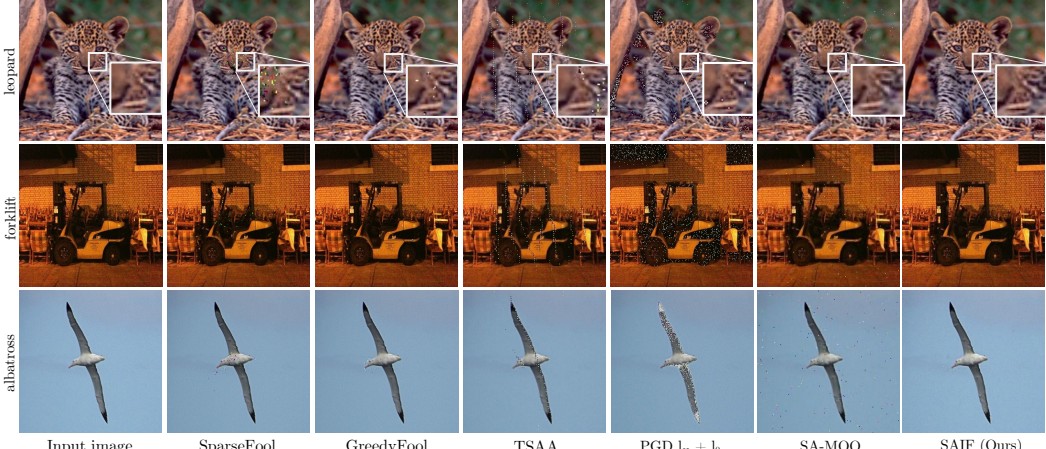

Figure 3: Visual results of **untargeted** attack on three images for $\epsilon = 255$, $k = 600$. Our method produces the most imperceptible adversarial examples despite the relaxed constraint on magnitude.

of ~98% for each $\epsilon$. For the same $\epsilon$, SAIF achieves $100\%$ ASR with **1.12%, 0.07%, and 0.037%** sparsity, respectively. Moreover, sparse attacks are more visually interpretable than $\ell_\infty$ attacks.

**Qualitative Results.** We include some examples of targeted adversarial examples using SAIF in Figure 2. Note that the perturbations **p** have been enhanced in all figures for visibility. Visually, the perturbations generated by SAIF are only slightly noticeable in regions with a uniform color palette/low textural detail, such as on the beige couch (see the zoomed-in segments of Figure 2). The other images are bereft of such regions and thus have negligible visible change. Moreover, we observe that the attack predominantly perturbs semantically meaningful pixels in the images.

For untargeted attacks, we present examples from all competing attack methods in Figure 3. We ease the magnitude constraint to allow all baselines to achieve some successful attacks at the same level of sparsity. It can be observed that all competing methods produce highly conspicuous perturbations around the face of the leopard, and on the outlines of the forklift and the albatross. PGD $\ell_\infty + \ell_0$, in particular, adds the most noticeable distortions to images. In contrast, SAIF attack produces adversarial examples that appear virtually identical to the clean input images.

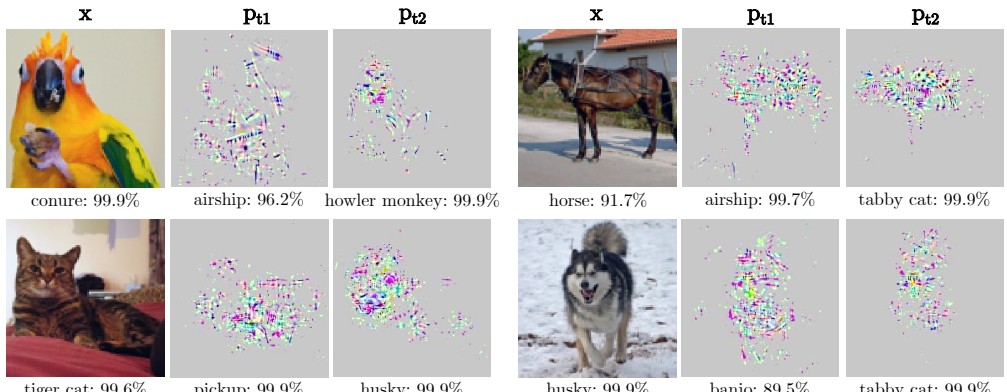

| x | $p_{t1}$ | $p_{t2}$ | x | $p_{t1}$ | $p_{t2}$ |

conure: 99.9%    airship: 96.2%    howler monkey: 99.9%    horse: 91.7%    airship: 99.7%    tabby cat: 99.9%

tiger cat: 99.6%    pickup: 99.9%    husky: 99.9%    husky: 99.9%    banjo: 89.5%    tabby cat: 99.9%

Figure 4: Impact of choice of target class on adversarial perturbations. We run targeted attacks for two target classes ($t_1$ and $t_2$) for each input. The perturbations have been enhanced for visualization.

**Perturbations and Interpretability** The sparse nature of perturbations allows us to study the interpretability of each attack (i.e. their correspondence with discriminative image regions). A similar evaluation is carried out in Xu et al. (2019), but they treat the saliency maps obtained from CAM (Zhou et al., 2016) as the ground truth. The saliency maps from CAM (and other existing methods) incur several failure cases. Therefore, we use the ImageNet bounding box annotations as a reliable baseline to analyze attack understandability.

We use ResNet50 as the target model and set $\epsilon = 255$ and $k = 2000$ for all attacks. The results are reported in Table 3. SAIF achieves the highest Loc. score among competing

| Attack | Loc. ↑ |
|---|---|
| SparseFool | 0.006 |
| GreedyFool | 0.001 |
| TSAA | 0.006 |
| SAIF (untargeted) | **0.126** |
| SAIF (targeted) | 0.118 |

Table 3: Quantitative evaluation of interpretability in terms of overlap with GT bounding boxes of ImageNet validation images. We use $\epsilon = 255, k = 200$ for all attacks in an untargeted setting unless indicated otherwise.

methods in both the targeted and untargeted attack setting. Note that SAIF performs better on this metric in the untargeted attack setting versus the targeted attack. This is intuitively valid since the untargeted objective only diminishes features of the true class. Whereas, targeted attacks introduce features to convince a DNN to predict the target class.

Moreover, from visual inspection (see Figure 4) it is observed that the nature of the target class also determines the sparse distortion pattern. In particular, attacking input images of an animate class towards another animate class ($t_2$) results in perturbations focused predominantly on the facial region in the image. The reverse is observed when attacking animate towards inanimate object classes ($t_1$), which typically modify the body of the subject in the image.

## 6.1 SPEED COMPARISON

Table 4 reports running times of all baselines. PGD $\ell_0 + \ell_\infty$ Croce & Hein (2019) runs the fastest but produces the most noticeable perturbations. Note that, although the inference time for GreedyFool (Dong et al., 2020) and TSAA (He et al., 2022) is $\leq 2$ seconds, these generator-based attacks require pre-training a generator for each target model (as well as each $\epsilon$ and target class for TSAA (He et al., 2022)). This incurs a significant computational overhead (>7 days on a single GPU), which offsets the faster optimization times for these approaches. SAIF attack relies solely on pre-trained classifiers, and is significantly faster than the existing state-of-the-art Homotopy-Attack (Zhu et al., 2021). Moreover, more efficient implementations of the FW LMO can further shorten running times, which we leave for future work.

| Attack | Time (sec) |
|---|---|
| SparseFool | 20 |
| GreedyFool | 1.7 |
| TSAA | 1.8 |
| SAPF | 1142 |
| Homotopy-Attack | 1500 |
| PGD $\ell_0 + \ell_\infty$ | **1.46** |
| SA-MOO | 56.07 |
| SAIF (ours) | 15 |

Table 4: Average running times on ImageNet

## 7 ABLATION STUDIES

We perform two sets of ablative experiments to highlight the significance of our design choices.

**Attack Sparsity.** To illustrate the importance of limiting the sparsity of attack using $\mathbf{s}$, we reformulate the problem to one constrained only over the perturbation magnitude. That is, we use the following objective for the untargeted attack:

$$D(\mathbf{p})_{adv} = \Phi(\mathbf{x} + \mathbf{p}, c) \tag{8}$$

$$\max_{\mathbf{p}} D(\mathbf{p})_{adv}, \quad \text{s.t.} \quad \|\mathbf{p}\|_\infty \leq \epsilon \tag{9}$$

Similarly, we reframe the targeted attack by optimizing for the objective $D(\mathbf{p})_{adv}$ where,

$$D(\mathbf{p})_{t,adv} = \Phi(\mathbf{x} + \mathbf{p}, t) \tag{10}$$

$$\min_{\mathbf{p}} D(\mathbf{p})_{t,adv}, \quad \text{s.t.} \quad \|\mathbf{p}\|_\infty \leq \epsilon \tag{11}$$

This is similar to Chen et al. (2020)'s attack method that uses Frank-Wolfe for optimization.

Following Table 1-2, we test the attack for various $\epsilon$ and report the results in Table 6 (Appendix B). It is observed that in the absence of a sparsity constraint, the attack distorts all image pixels regardless of the constraint on magnitude. Moreover, such spatially 'contiguous' perturbations are visible even at magnitudes as low as $\epsilon = 10$ (see Figure 5, Appendix B). By constraining the sparsity for SAIF attack, we ensure the adversarial perturbations stay imperceptible even at $\epsilon = 255$, at which the non-sparse attack completely obfuscates the image.

**Loss formulation.** We also experiment with different losses, mainly the $\ell_2$-attack proposed by Carlini & Wagner (2017), but observe a decline in attack success (see Table 5. We choose $\epsilon$ and $k$ for which SAIF achieves 100% ASR in Table 1.). A possible explanation for this behavior is that the loss formulation tries to increase the target class probability too aggressively, which makes the simultaneous optimization of $\mathbf{s}$ and $\mathbf{p}$ difficult. We observe that this yields solutions closer to the constraint boundaries, increasing the attack visibility.

| Constraints | ASR |
|---|---|
| $\epsilon = 255, k = 600$ | 97.78% |
| $\epsilon = 100, k = 600$ | 98.30% |
| $\epsilon = 10, k = 3000$ | 88.97% |

Table 5: ASR for targeted attacks on Inception-v3 when cross-entropy is replaced with $\ell_2$-attack loss (Carlini & Wagner, 2017).

## 8 CONCLUSION

In this work, we propose a novel adversarial attack, 'SAIF', by jointly minimizing the magnitude and sparsity of perturbations. By constraining the attack sparsity, we not only conceal the attacks but also identify the most vulnerable pixels in natural images. We use the Frank-Wolfe algorithm to optimize our objective and achieve effective convergence, with reasonable efficiency, for large natural images. We perform comprehensive experiments against state-of-the-art attack methods and demonstrate the remarkably superior performance of SAIF under tight magnitude and sparsity budgets. Our method also outperforms existing methods on a quantitative metric for interpretability and provides transparency to visualize vulnerabilities of DNNs.

## DISCUSSION

**Social Impact** Adversarial attacks expose the fragility of DNNs. Our work aims at demonstrating that a highly imperceptible adversarial attack can be generated for natural images. This provides a new benchmark for the research community to test the robustness of the learning algorithms. A straightforward defense strategy can be using the adversarial examples generated by SAIF for adversarial training. We leave more advanced solutions for further exploration.

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

## APPENDIX

## A  FRANK WOLFE ALGORITHM

The Frank-Wolfe algorithm (FW) 2 (Frank et al., 1956) is a first-order, projection-free algorithm for optimizing a convex function $f(\mathbf{x})$ over a convex set $\mathcal{X}$. The key advantage of FW is that the iterates $\mathbf{x}_t$ always remain feasible ($\mathbf{x}_t \in \mathcal{X}$) throughout the optimization process.

Frank-Wolfe is a projection-free method since it conservatively solves a linear approximation of the objective $f$ over the feasible region $\mathcal{X}$. The set $\mathcal{X}$ may be described as the convex hull of a (possibly infinite) set of atoms $\mathcal{A}$. In the case of the $\ell_1$ ball ($\mathcal{X} = \{\mathbf{x} \in \mathbb{R}^n \mid \|\mathbf{x}\|_1 \leq \tau\}$), these atoms may be chosen as the $2n$ unit vectors, i.e., $\mathcal{A} = \{\pm \mathbf{e}_j, \ j = 1 \dots n\}$.

---

**Algorithm 2:** Frank-Wolfe Algorithm

---

**Input:** Objective $f$, convex set $\mathcal{X}$, Maximum iterations $T$, stepsize rule $\eta_t$
**Output:** Final iterate $\mathbf{x_T}$

1   $\mathbf{x_0} = \mathbf{0}$
2   **for** $t = 0 \dots T - 1$ **do**
3      $\mathbf{a_t} = \arg\min_{\mathbf{a} \in \mathcal{X}} \langle \nabla f(\mathbf{x_t}), \mathbf{a} \rangle$
4      $\mathbf{x_{t+1}} = \mathbf{x_t} + \eta_t(\mathbf{a_t} - \mathbf{x_t})$
5   **end**

---

## B  ABLATION STUDIES - RESULTS

The results for ablating over the attack sparsity are reported in Table 6. Figure 5 shows examples of samples attacked with and without a sparsity constraint, under the same perturbation magnitude constraints $\epsilon$.

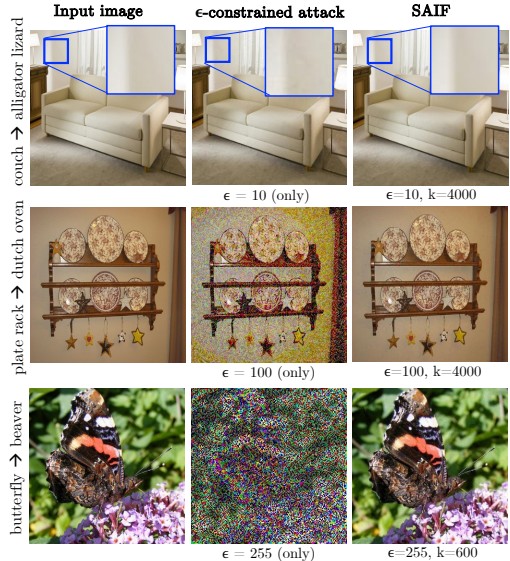

| Attack Type | Dataset | ASR ↑ | $\|\mathbf{p}\|_0/m \downarrow$ |
|---|---|---|---|
| untargeted | $\epsilon = 255$ | 100.0 | 1.0 |
| | $\epsilon = 100$ | 100.0 | 1.0 |
| | $\epsilon = 10$ | 100.0 | 1.0 |
| targeted | $\epsilon = 255$ | 100.0 | 1.0 |
| | $\epsilon = 100$ | 100.0 | 1.0 |
| | $\epsilon = 10$ | 100.0 | 1.0 |

Figure 5: Exploring the significance of sparsity of the adversarial attack. When the sparsity is not constrained (middle column), perturbations of very small magnitude ($\epsilon = 10$) are noticeable and completely distort the image for larger $\epsilon$. On the contrary, SAIF (third column) stays imperceptible at higher magnitudes as well.

Table 6: Quantitative evaluation of optimizing SAIF without a sparsity constraint for Inception-v3 on ImageNet. For each perturbation magnitude, the attack distorts all the pixels in the image leading to high perceptibility.

## C   RESULTS ON ADDITIONAL DATASET

For a comprehensive evaluation of our attacks, we also test our approach on a smaller dataset.

### C.1   DATASET AND MODEL

We test SAIF and the existing sparse attack algorithms on the CIFAR-10 dataset Krizhevsky et al. (2009). The dataset comprises of $[32 \times 32]$ RGB images belonging to 10 classes. We evaluate all algorithms on 10,000 samples from the test set.

We attack VGG-16 (Geifman, 2019) trained on CIFAR-10, having an accuracy of 92.32% on clean images.

### C.2   QUANTITATIVE RESULTS

We run all attacks for a range of $\epsilon$ and $k$, and report the results in Tables 7,8.

Table 7 reports the ASR for the untargeted attack. SAIF consistently outperforms SparseFool (Modas et al., 2019) and GreedyFool (Dong et al., 2020) on all sparsity and magnitude constraints. Similar results are obtained for targeted attacks, reported in Table 8. The target classes are randomly chosen in all experiments.

### C.3   QUALITATIVE RESULTS

We provide several examples of adversarial examples produced by SAIF and competing algorithms. Figure 6 shows samples obtained by untargeted attacks on VGG-16 trained on CIFAR-10. It is observed that SAIF consistently produces the most imperceptible perturbations.

| $\|\mathbf{p}\|_\infty \leq \epsilon$ | Attacks | VGG-16 | | | | |
|---|---|---|---|---|---|---|
| | | Sparsity '$k$' | | | | |
| | | 1 | 2 | 5 | 10 | 20 |
| | SparseFool (Modas et al., 2019) | 10.78 | 18.56 | 38.32 | 63.67 | 85.23 |
| $\epsilon = 255$ | GreedyFool (Dong et al., 2020) | 0.00 | 0.00 | 24.75 | 69.26 | 85.83 |
| | SAIF (Ours) | **91.20** | **92.87** | **94.46** | **96.35** | **100.0** |
| | | 5 | 10 | 15 | 20 | 30 |
| | SparseFool (Modas et al., 2019) | 30.54 | 51.50 | 65.47 | 74.45 | 86.43 |
| $\epsilon = 100$ | GreedyFool (Dong et al., 2020) | 25.95 | 55.09 | 67.26 | 73.65 | 86.43 |
| | SAIF (Ours) | **91.89** | **92.84** | **93.57** | **96.74** | **97.80** |
| | | 30 | 40 | 50 | 60 | 100 |
| | SparseFool (Modas et al., 2019) | 19.36 | 24.75 | 27.94 | 31.94 | 44.51 |
| $\epsilon = 10$ | GreedyFool (Dong et al., 2020) | 33.93 | 39.92 | 45.91 | 51.10 | 66.27 |
| | SAIF (Ours) | **90.32** | **91.57** | **92.14** | **92.65** | **94.14** |

Table 7: Quantitative evaluation of **untargeted** attack on CIFAR-10. We report the ASR for varying constraints on sparsity '$k$' and $\ell_\infty$-norm of the magnitude of perturbation '$\epsilon$'.

| $\|\mathbf{p}\|_\infty \leq \epsilon$ | Attacks | VGG-16 | | | | |
|---|---|---|---|---|---|---|
| | | Sparsity '$k$' | | | | |
| | | 1 | 2 | 5 | 10 | 20 |
| $\epsilon = 255$ | GreedyFool (Dong et al., 2020) | 0.00 | 0.00 | 2.79 | 13.17 | 29.94 |
| | SAIF (Ours) | **12.36** | **12.83** | **27.02** | **44.05** | **61.10** |
| | | 5 | 10 | 15 | 20 | 30 |
| $\epsilon = 100$ | GreedyFool (Dong et al., 2020) | 2.20 | 9.58 | 14.97 | 16.97 | 30.74 |
| | SAIF (Ours) | **13.37** | **21.03** | **26.27** | **36.18** | **51.43** |
| | | 30 | 40 | 50 | 60 | 100 |
| $\epsilon = 10$ | GreedyFool (Dong et al., 2020) | 3.99 | 5.19 | 6.99 | 8.98 | 16.17 |
| | SAIF (Ours) | **5.46** | **13.35** | **18.13** | **22.25** | **29.26** |

Table 8: Quantitative evaluation of **targeted** attack on CIFAR-10. We report the ASR for varying constraints on sparsity '$k$' and $\ell_\infty$-norm of the magnitude of perturbation '$\epsilon$'.

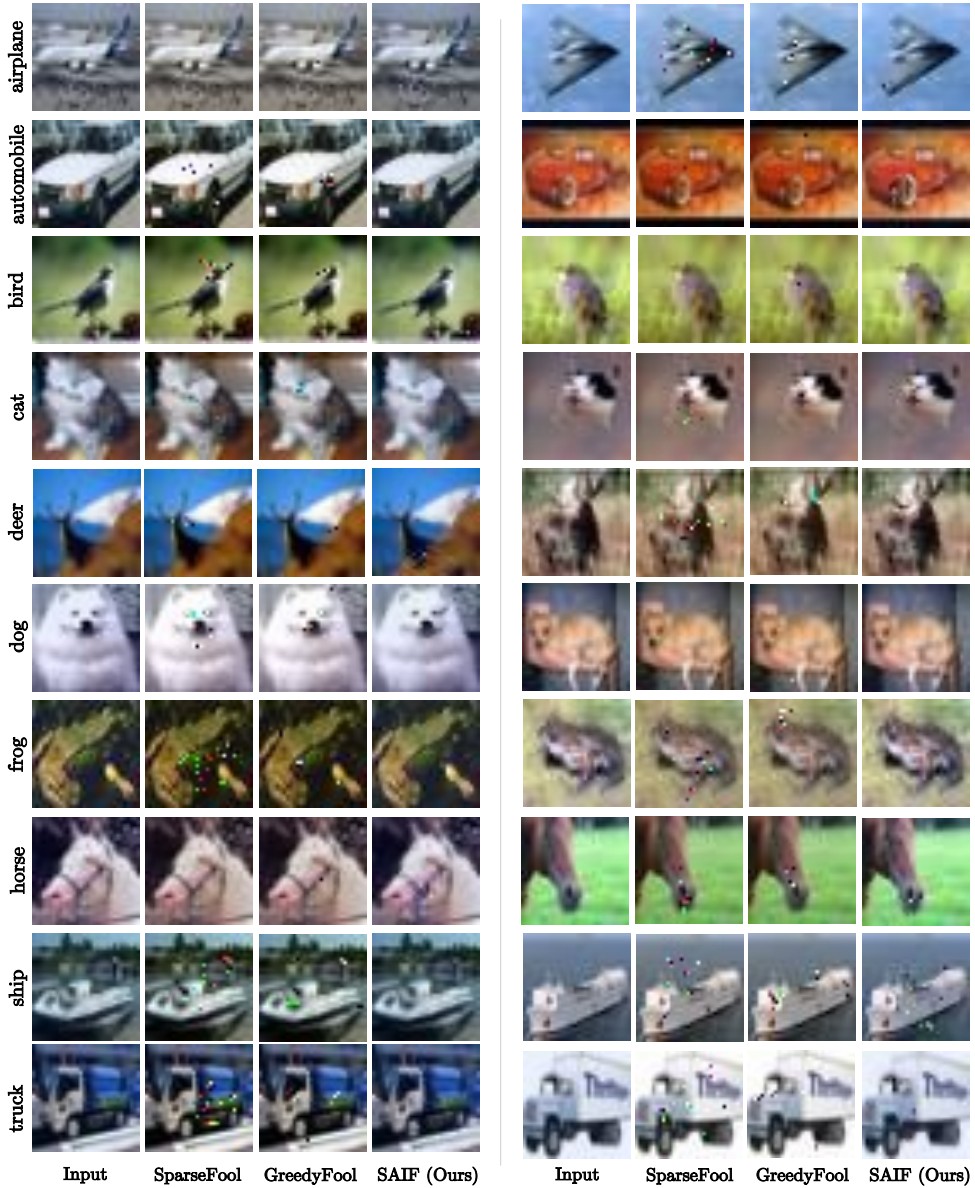

Figure 6: Untargeted adversarial examples obtained from SAIF and competing attack algorithms on CIFAR-10. The attacked classifier is VGG-16

