# OpenReview forum: "SAIF: Sparse Adversarial and Imperceptible Attack Framework"
_ICLR.cc/2024/Conference — ICLR 2024 Conference Withdrawn Submission_

### Official Review · Reviewer_a5or · 2023-11-02

**Soundness:** 3 good
**Presentation:** 3 good
**Contribution:** 3 good
**Rating:** 6
**Confidence:** 3

**Summary:**

The paper introduces SAIF, a novel adversarial attack framework that crafts highly imperceptible and sparse perturbations to expose vulnerabilities in neural network classifiers. Utilizing the Frank-Wolfe algorithm, SAIF optimizes for bounded magnitude and sparsity, achieving a convergence rate of $O\left(\frac{1}{\sqrt{T}}\right)$. Empirical results demonstrate its superiority, significantly outperforming state-of-the-art sparse attack methods on ImageNet and CIFAR-10 datasets. This work not only highlights the susceptibility of classifiers to subtle perturbations but also contributes to the broader understanding of neural network robustness, providing a valuable tool for evaluating and enhancing adversarial resilience.

**Strengths:**

The authors use an elegant Frank-Wolfe algorithm for the simultaneous optimization of adversarial perturbations. An $\ell_1$ convex surrogate is used for sparsity. SAIF demonstrates a sound technical approach, achieving a rapid convergence rate and ensuring the generation of adversarial examples with bounded magnitude and sparsity. The SAIF method outperforms state-of-the-art sparse attack methods on major datasets like ImageNet and CIFAR-10, especially when fewer pixels are allowed to be perturbed. SAIF also provides interpretable adversarial examples.

**Weaknesses:**

* The $O(1/\sqrt{T})$ convergence analysis is very interesting. But I was wondering how this number reflects the real running time in practice. For example, what are the convergence rates for existing methods in Table 4?
* The models tested are un-defended. The authors may need to test some robust models. Even the 'regular' robust models adversarially trained with small $\epsilon$ values and without sparsity constraints are helpful. Another way is to train the model with SAIF on the fly, to see if it performs better under SAIF.

**Questions:**

* Given the perturbations' interpretability, I think it would be good if the authors could provide some transferability analysis across different model structures on SAIF, and compare it with existing works.

---

### Official Review · Reviewer_Taxv · 2023-11-02

**Soundness:** 3 good
**Presentation:** 2 fair
**Contribution:** 2 fair
**Rating:** 3
**Confidence:** 5

**Summary:**

The paper proposed Sparse Adversarial and Imperceptible Attack Framework (SAIF) to generate adversarial images while simultaneously controlling both the $\ell_\infty$- and $\ell_0$-norm of the corresponding perturbations. SAIF relies on Frank-Wolfe algorithms to optimize targeted and untargeted attacks. In the experiments on ImageNet, SAIF is shown effective in several setups and in comparison to existing attacks. Finally, the paper shows that sparse attacks might be more interpretable than dense ones, and find salient pixels in the image.

**Strengths:**

- The proposed algorithm to simultaneously optimize the dense perturbation and sparse mask seems overall novel.

- in the experiments, SAIF is shown to outperform existing attacks on both targeted and untargeted attacks.

**Weaknesses:**

- It is not clear how the $\ell_0$-norm constrain is enforced: in Line 7 of Alg. 1, the sparsity of $s_t$ seems to decrease compared to $s_{t-1}$ (unless $\eta_t=1$ or $s_{t-1}$ and $z_t$ have the non-zero components and the same position). Moreover, the presentation in general could be more clear, e.g. some notation is used before being introduced.

- The case with $\epsilon=255$ is equivalent to a standard $\ell_0$-attack: then, I think a comparison to existing attacks (e.g. [A, B, C]) in this threat model should be included. Moreover, some of these attacks, e.g. [C], could be adapted to include $\ell_\infty$ bounds in a quite straightforward way, providing further baselines for the $\epsilon < 255$ setups.

- Some of the attacks used in the comparison, e.g. PGD $\ell_0 + \ell_\infty$, use sparsity constraints in the pixel space, while, if I understand it correctly, SAIF counts color channels individually. It's not clear how this difference, which would make the results of different attacks hardly comparable, is handled. Moreover, PGD $\ell_0 + \ell_\infty$ is flagged as a black-box attack, while it is a white-box method. Finally, the parameters, e.g. number of iterations, used for the existing methods are not detailed.

- The attacks are compared only on ResNet-50 and Inception-v3. Including more recent (and effective) architectures, and maybe adversarially trained models, would make the experimental evaluation more complete.

- The ablation study about removing the sparsity constrain (Sec. 7) doesn't seem to add any valuable insight into the proposed method, since it is known that $\ell_\infty$-bounded attacks perturb (almost) all pixels (which is also suggested by the update step in Eq. (5)).

[A] https://arxiv.org/abs/2006.12834 \
[B] https://arxiv.org/abs/2102.12827 \
[C] https://arxiv.org/abs/2106.01538

**Questions:**

See weaknesses above.

Overall, the paper lacks, at the moment, clarity in the presentation of the algorithm and experiments. Moreover, relevant baselines might be missing.

---

### Official Review · Reviewer_9tC9 · 2023-11-07

**Soundness:** 2 fair
**Presentation:** 2 fair
**Contribution:** 2 fair
**Rating:** 5
**Confidence:** 4

**Summary:**

The paper considers the generation of a per-input white-box adversarial attack on vision classifiers constrained by $L_0$ and $L_{\infty}$ bounds simultaneously. To efficiently solve the resulting optimization problem, the authors leverage the classical Frank-Wolfe (FW) algorithm. The experimental evaluation is performed on the ImageNet and CIFAR10 datasets using the attack success rate and attack saliency metrics. The results show the promise of the approach.

**Strengths:**

* The use of the FW algorithm for crafting stealthy adversarial examples is novel and intuitive.

* Within the proposed threat model, the evaluation results are strong.

**Weaknesses:**

* I did not find a practical motivation behind the proposed threat model. Sparse and low-magnitude attacks can indeed be hard to detect. However, to create the SAIF attacks, the attacker needs unrealistic powers: (i) it has white-box access to the model parameters, (ii) it knows the exact inputs processed by the model in advance, and (iii) it can also make the target model perceive the examples perfectly. The runtime of the SAIF algorithm is also slow so that makes real-time attack generation difficult. If the goal is to study the theoretical robustness of the model as with other unrealistic attacks, then stealthiness and low magnitude together are not really necessary and constraining the adversary by $L_{\infty}$ or $L_{0}$ already gives a stronger measure of robustness than the threat model considered here.

* Vision transformers have been shown to be more robust in literature (https://arxiv.org/abs/2105.07581) than the Inception and ResNet models considered here. It is not clear if the attack is effective on these more robust models.

* I found the technical details related to FW hard to digest as the authors assume familiarity with the details of FW. For example, computing $g(x_t)$ in section 3 involves maximizing over the set $\mathcal{A}$, but this is never defined. Similarly, the text in Section 4 suddenly starts talking about the undefined vectors $z_t, v_t$ and conditional gradients.

**Questions:**

* The authors should either show practical attacks with SAIF or motivate why a theoretical study of robustness within the SAIF model is necessary. Stealthy attacks may have applications for captcha generation (https://arxiv.org/abs/2101.02483) but that scenario will require evaluation against extra baselines.

* Can the FW method be adapted for more diverse constraints? e.g., rotation+ $L_0$ + snow?

* I like the study about the interpretability of the perturbations but the authors did not motivate any end-to-end benefits of that to a practitioner.